# Allergic Rhinitis: Pathophysiology and Treatment Focusing on Mast Cells

**DOI:** 10.3390/biomedicines10102486

**Published:** 2022-10-05

**Authors:** Yara Zoabi, Francesca Levi-Schaffer, Ron Eliashar

**Affiliations:** 1Department of Pharmacology and Experimental Therapeutics, School of Pharmacy, Faculty of Medicine, Hebrew University, Jerusalem 9112002, Israel; 2Department of Otolaryngology/HNS, Hadassah-Hebrew University Medical Center, Jerusalem 91120, Israel

**Keywords:** allergy, allergic rhinitis, rhinosinusitis, mast cells, pharmacologic therapy, biologics

## Abstract

Allergic rhinitis (AR) is a common rhinopathy that affects up to 30% of the adult population. It is defined as an inflammation of the nasal mucosa, develops in allergic individuals, and is detected mostly by a positive skin-prick test. AR is characterized by a triad of nasal congestion, *rhinorrhea,* and sneezing. Mast cells (MCs) are innate immune system effector cells that play a pivotal role in innate immunity and modulating adaptive immunity, rendering them as key cells of allergic inflammation and thus of allergic diseases. MCs are typically located in body surfaces exposed to the external environment such as the nasal mucosa. Due to their location in the nasal mucosa, they are in the first line of defense against inhaled substances such as allergens. IgE-dependent activation of MCs in the nasal mucosa following exposure to allergens in a sensitized individual is a cardinal mechanism in the pathophysiology of AR. This review is a comprehensive summary of MCs’ involvement in the development of AR symptoms and how classical AR medications, as well as emerging AR therapies, modulate MCs and MC-derived mediators involved in the development of AR.

## 1. Introduction

Allergic rhinitis (AR) is a common allergic inflammatory rhinopathy that affects up 20 to 30% of adults and up to 40% of children worldwide [1]. Rhinitis is defined as inflammation of the lining of the nose; AR is the most widespread form of noninfectious rhinitis. Similar to other allergic diseases, AR is a result of an inordinate response of the immune system of a sensitized individual to innocuous substances known as allergens. [1,2].

AR is classically characterized by ongoing symptoms of *rhinorrhea*, nasal congestion and blockage, sneezing, and/or itching of the nose. These symptoms may significantly impact the patients’ quality of life, often interfering with sleep and contributing to poor academic and work performance [1].

Diagnosis of AR is usually made clinically when an individual experiences one or more of the hallmark symptoms mentioned above in response to allergen exposure. AR is classified as seasonal when the symptoms are experienced seasonally (i.e., pollen season) and perennial when the symptoms are experienced year-round. Another classification method is based on the length and recurrence of the symptoms. Intermittent AR (IAR) is defined as symptoms persisting for less than 4 days/week or less than 4 weeks, while persistent AR (PAR) is defined as symptoms lasting for more than 4 days/week for at least 4 weeks [3].

To confirm the diagnosis, sensitization to allergens is assessed by an allergy skin-prick test or an in vitro antigen-specific IgE test. Clinical allergy is evidenced by active symptoms upon allergen exposure in a sensitized individual. A unique condition is local allergic rhinitis, which is characterized by local IgE production in the nasal tissues whilst systemic allergy testing is negative [3].

Management of AR symptoms starts with allergen (i.e., house dust mite, animal dander, pollen, etc.) avoidance. AR symptoms are mainly controlled via pharmacotherapeutic agents. These include antihistamines, intranasal corticosteroids, leukotriene receptor antagonists (LTRAs), cromolyn sodium, biologics, and immunotherapy [4].

As elaborated in detail later in this review, exposure to inhaled allergens in a sensitized individual starts the cascade leading to the AR symptoms via activation of MCs lining the nasal mucosa through crosslinking of antigens to a specific IgE antibody bound to the surface of MCs, resulting in the release of MC mediators. The latter contribute to the immediate allergic response and to further recruitment and infiltration of immune cells to the site of inflammation, including eosinophils, B cells, and T cells. The primary T-cell response in AR is mediated by T-helper (Th2) cells. Th2 cells secrete cytokines such as interlukin-4 (IL-4) and interlukin-13 (IL-13), which drive further IgE production. This *milieu* of inflammatory cells and mediators contributes to the late and chronic phase response of allergic inflammation [5,6].

In addition to AR, MCs’ role is established in the pathogenesis of frequent comorbidities characterized by allergic inflammation such as atopic dermatitis (AD) and allergic asthma. To promote allergic inflammation, MCs interact with other inflammatory cells via soluble mediators and cell–cell contact. One of these interactions occurs through the allergic effector unit (AEU), where CD48 on MCs crosstalk with a 2B4 receptor on eosinophils, promoting an inflammatory outcome [7,8,9].

A differential diagnosis of AR includes other types of sinonasal diseases because symptoms may overlap. In fact, more than one type of rhinitis may coexist at the same time [3]. An example is rhinosinusitis, another rhinopathy that involves inflammation of the paranasal sinuses and is characterized by the presence of postnasal drip, facial pressure, and a reduction or loss in the ability to smell [1,10]. Chronic rhino sinusitis (CRS) is defined as the persistent symptoms of nasal obstruction, *rhinorrhea*, hyposmia, and facial pain for more than 12 weeks in combination with inflammatory signs confirmed by nasal endoscopy or by a computed tomography scan. Phenotypically, CRS is divided into two groups based on the presence or absence of nasal polyps: CRS with nasal polyps (CRSwNP) and CRS without nasal polyps (CRSsNP) [3,10,11]. Emerging evidence further divides CRS into different endotypes within each phenotypic category to better adjust treatment and management protocols [12]. The endotype of CRSwNP in most patients is of the Type 2 inflammation and typically shows tissue eosinophils and MCs [10], a common feature bridging CRSwNP and AR. 

Another condition associated with CRSwNP is aspirin-induced respiratory disease (AERD), which is defined as a triad of adult onset asthma, CRSwNP, and anaphylactic response to COX-1 inhibitors (i.e., NSAIDs). Activation of MCs by classical (IgE-mediated) and nonclassical pathways is crucial in the pathophysiology of AERD due to the secretion of a diverse array of inflammatory mediators that activate different Type 2 inflammatory pathways involved in AERD pathogenesis [13].

As demonstrated above, MCs are essential players in the development of allergic diseases, including AR. 

## 2. Mast Cells’ Role in Allergic Inflammation

The allergic reaction consists of an early phase mediated by the release of inflammatory mediators from preformed granules of MCs and a late phase characterized by the influx of inflammatory cells, which are mostly modulated by MCs and their secreted mediators [10,11].

MCs are located in the major body surfaces exposed to the external environment. This includes the skin, where connective tissue MCs are primarily found; the epithelium of the gastrointestinal tract; and the respiratory tract, where mainly mucosal MCs are found. The content of the preformed granules divides MCs into two phenotypic subtypes: mucosal MCs producing only tryptase and connective tissue MCs producing tryptase, chymase, and carboxypeptidase [14].

Due to their strategic positioning in the exposed body linings and the stored granules laden with preformed mediators, MCs may respond to invading stimuli more rapidly than other tissue-resident immune cells [14,15], thus playing a cardinal role as the host’s first-line of defense against invading organisms [16].

FcεRI is a high-affinity IgE receptor found on the surface of MCs and basophils. Allergen crosslinking of the IgE-bound FcεRI receptors results in the release of preformed and newly synthesized mediators in a phasic fashion. A few seconds after activation, MCs degranulate and release the stored content of the preformed granules, which contain histamine, heparin, proteases, and TNFα [14].

In addition, arachidonic acid is freed from membrane phospholipids and de novo synthesized arachidonic acid–lipid metabolites—mainly prostaglandin D2, leukotriene LTC4/D4/E4, and LTB4—are also promptly released as part of the immediate MC response. [17,18] Later, a vast array of cytokines, chemokines, and growth factors are synthesized and released [14,16]. After degranulation, MCs can resynthesize their preformed mediators. 

MCs promote the trafficking of immune cells to the site of infection or inflammation, making them an essential initiator of the inflammatory response. MCs recruit circulating eosinophils through direct secretion of eosinophil-attracting chemokines (i.e., eotaxins) and indirectly via secretion of histamine, inducing eotaxin secretion by endothelial cells [16].

MCs and eosinophils are the key effector cells in allergy and communicate with each other in a bidirectional manner in what is defined as the “Allergic Effector Unit”. This crosstalk is mediated by physical cell–cell contact through cell surface receptors/ligands and by released mediators, including specific granular factors, chemokines, cytokines, and their respective ligands. In the physical cell–cell crosstalk, CD48 on MCs and its high-affinity ligand, the 2B4 receptor on eosinophils, form a pivotal couple, initiating and maintaining this proinflammatory crosstalk [8,19].

Through complex cellular processes, MCs can influence the induction, amplitude, and function of the adaptive immune response. Via histamine secretion, MCs increase vessel permeability, aiding in the recruitment of adaptive cells to the site of inflammation [16].

## 3. MCs’ Link to the Adaptive Immune Response

Dendritic cells (DCs) are antigen-presenting cells (APCs) identified by their high expression of major histocompatibility complex (MHC) class II molecules (MHC-II). In the tissue, they detect homeostatic imbalances and process antigens for presentation to T lymphocytes, linking the innate and the adaptive immune responses with one another [20]. MCs secrete CCL20, inducing recruitment of DC precursors from the blood into the inflamed site [21]. In addition, MCs modulate DC activity by promoting tissue edema (due to increased vessel permeability mediated by histamine), which is important for the motility of the DCs, supporting their subsequent migration toward the draining lymph nodes and thus inducing antigen-specific adaptive immune responses [16].

## 4. MCs’ Role in AR Initiation, Progression, and Modulation

In the nasal mucosa, APCs are located in paracellular and intercellular channels adjacent to the basal epithelial cells. APCs process inhaled allergen-derived proteins deposited in the nasal mucosa and present them to naïve CD4^+^ T lymphocytes (Th0s) in the draining lymph nodes. Th0s are activated and transformed into T-helper 2 CD4^+^ lymphocytes (Th2s) under interleukin-4 (IL-4) cytokine stimulation. Th2 cells, together with ILC2, release IL-4, IL-5, and IL-13 to initiate the adaptive immune response. IL-4 and IL-13 then stimulate specific B-cell lymphocytes to differentiate into antibody-producing plasma cells [18]; IL-4 and IL-13 are therefore required to generate and sustain IgE production [21,22].

Additionally, exposure to allergens induces structural cells in the nasal mucosa (e.g., epithelial cells) to release cytokine alarmins including thymic stromal lymphopoietin (TSLP), IL-25, and IL-33, which in turn promote the downstream production of proinflammatory Type 2 inflammatory cytokines by effector immune cells [18,23,24].

As mentioned earlier, eosinophils engage MCs in a physical and soluble crosstalk via the AEU. MCs and eosinophils produce mediators that enhance the activation status of both cells [25]. Humanized anti-IL5 antibodies (e.g., **mepolizumab**) and anti-IL5R antibodies (e.g., **benralizumab**) that interfere with the IL-5 pathway [26] may thus affect the interaction of MCs and eosinophils within the AEU, influencing the course of MC-related diseases [25].

TSLP, which is secreted by structural cells in the nasal mucosa, binds the TSLP receptor (TSLPR) expressed by MCs, inducing their release of various cytokines and chemokines (e.g., IL-5, IL-13, and CCL1) [24]. Due to TSLP’s role in driving Type 2 inflammation, clinical studies have been investigating the monoclonal antibody targeting TSLP (**tezepelumab)** in asthma and atopic dermatitis [27].

## 5. Phases of Allergic Inflammation

Histamine, one of the preformed mediators in MCs, is known to be a major factor of the acute allergic response. Its release from MCs initiates the immediate (or early) phase response, typically occurring within minutes after allergen exposure and lasting for 1–2 h after exposure. Histamine binds to four types of G-coupled receptors. In the nasal mucosa, histamine induces activation of H1 receptors on sensory nerves of the afferent trigeminal system, which in turn transmit signals to the central nervous system (CNS), causing symptoms of itching and motor reflexes; i.e., sneezing. Via activation of sensory and parasympathetic nerves, histamine stimulates mucous glands to secrete a watery discharge, which manifests clinically as *rhinorrhea*. Activation of H1 and H2 receptors of nasal blood vessels by histamine contributes to increased vascular permeability and vasodilation, leading to symptomatic nasal congestion and enhanced leukocyte recruitment to the inflamed nasal mucosa [2,18,28,29].

In addition, MCs release growth factors (e.g., fibroblast growth factor-2 (FGF-2) and vascular endothelial growth factor (VEGF)), which are angiogenic and increase vasodilation and vascular permeability in the nasal walls, leading to enhanced inflammatory cell infiltration, local edema, and swelling of the nasal mucosa, which contribute to the clinical symptoms of nasal congestion and watery *rhinorrhea* in AR [11]. 

During the early and late phase responses, MCs release cytokines and chemokines to attract additional inflammatory cell types to the nasal mucosa, including neutrophils and eosinophils, group 2 innate lymphoid cells (ILC2), and Th2 cells. An influx of these cells in the nasal mucosa characterizes the late-phase response, which typically occurs within 5 h following the initial allergen exposure and lasts up to 24 h. The late-phase response is complex due to the secretion of various cytokines and chemokines from an array of migratory cells that interact together to sustain the inflammation and prolong its symptoms through the release of additional cytokines [18] (Figure 1).

Cysteinyl leukotrienes and PGD2 released from MCs act by increasing the vascular permeability in the nasal mucosa [30,31,32]. In addition, they promote the recruitment and activation of ILC2 cells to the nasal mucosa [33]. ILC2, which also are resident cells in mucosal linings [28], are capable of releasing large amounts of Th2 cytokines within the mucosal nasal tissue, thus further sustaining the inflammatory response. Activated Th2 cells contribute to allergic inflammation mainly via the release of IL-5, which activates and further recruits eosinophils to the nasal mucosa. Eosinophils release superoxide anions, hydrogen peroxide, the eosinophil cationic protein (ECP), eosinophil-derived neurotoxin (EDN), eosinophil peroxidase (EPO), and the major basic protein (MBP), contributing to the damage to the nasal epithelium [18].

Resolution of inflammation can be expected after the two phases of allergy (early and late phases). However, due to the continuous exposure to the allergen(s) and/or the severity of the disease or atopy, allergic inflammation may evolve into a chronic phase when the allergic reaction fails to resolve (Figure 1). During the late and chronic phases of allergic inflammation, MCs and eosinophils are primarily involved and abundantly coexist in the inflamed tissue [8].

## 6. CD48 in AR and CRSwNP

CD48 is the high-affinity ligand for 2B4, an activating receptor expressed notably on eosinophils. The interaction between CD48 and 2B4 on MCs and eosinophils is an important mechanism by which MCs and eosinophils affect and communicate with one another within the AEU.

Our group demonstrated in a recent immunohistochemical study that MC infiltration in nasal polyps (NPs) obtained from CRSwNP patients was higher in allergic patients in comparison to nonallergic patients. In addition, membranal CD48 (mCD48) expression on eosinophils infiltrating nonallergic asthmatic NPs was highly elevated in comparison to allergic asthmatic polyps, allergic nonasthmatic polyps, and nonallergic nonasthmatic polyps. Similarly, mCD48 and the expression of its high-affinity ligand m2B4 on eosinophils from enzymatically digested NPs were significantly higher in nonallergic asthmatics in comparison to allergic asthmatics, possibly indicating that allergy attenuates mCD48 expression on eosinophils [19].

Recent findings demonstrated that sCD48 (the soluble form of CD48) levels were significantly higher in an intermittent allergic rhinitis group (IAR) of patients in the pollen season compared to their levels (in the same group of patients) not during pollen season. These results suggested that sCD48 may be a biomarker to the exacerbation phase in patients with IAR and that CD48 could be plausibly implicated in the pathogenesis of IAR [34].

## 7. Differential Gene Expression Profiles in AR Patients

A recent study aimed to compare the immune gene expression profiles in the nasal mucosa and peripheral blood samples between adults with AR and healthy controls. The results showed a significant upregulation of allergy-related genes in the nasal mucosa samples obtained from the AR group in comparison to the control group. Notably, chemokines genes, including CCL17, CCL26, and TPSAB1 (which encodes tryptase) were upregulated. Counts of differentially expressed genes (DEGs) in the nasal mucosa samples positively correlated with eosinophil and dust-mite-specific immunoglobulin E (IgE) counts in the blood. The study concluded that distinct gene expression profiles in the blood and nasal mucosa samples were observed between AR patients and healthy controls. The results also provided evidence for the close interaction between the local site and systemic immunity. The genes identified in this study contributed to the current knowledge on AR pathophysiology and may serve as biomarkers to evaluate the effectiveness of treatment regimens or as targets for therapeutic development [35].

## 8. Overview of AR Treatment

The allergic reaction in AR is characterized by an interplay between inflammatory cells infiltrating the nasal mucosa and released mediators that provokes the main symptoms of AR. Available therapeutic strategies in allergic diseases including AR, asthma, and atopic dermatitis are aimed largely at targeting MCs to suppress the allergic process [15].

Once a diagnosis of AR has been established, the standard of care includes a treatment plan that considers the severity of the disease; the presence of concomitant allergic diseases; and most importantly, a shared decision-making process that focuses on the patient’s preferences. The standard treatment algorithm for AR begins with allergen avoidance, including limiting exposure to relevant allergens, maintaining a humidity of less than 40% at home to prevent dust mites and mold, and use of high-efficiency particulate air (HEPA) filters to remove allergens from the inhaled air. If the symptoms persist despite avoidance strategies, pharmacologic options are discussed with the patient. The first-line approach is to use symptomatic treatment starting with antihistamines (AHs) and intranasal glucocorticosteroids (GCSs). For more severe disease, topical anti-inflammatory therapy is considered, and lastly, allergen-specific immunotherapy (AIT) may be employed, thus providing a bonus benefit due to its disease-modifying effects. Still, none of these modalities is optimal [36].

## 9. AR Pharmacotherapy Specifically Targeting MCs and Their Pathways (Table 1)

***Antihistamines*** are considered the first line of treatment for symptomatic AR and elicit their effects by blocking the histamine receptors expressed on nasal mucosa cells, thus preventing their activation by histamine secreted by MCs. Currently available oral antihistamines for the treatment of AR include first-generation antihistamines (diphenhydramine, chlorpheniramine, and hydroxyzine) and second-generation antihistamines (fexofenadine, loratadine, desloratadine, and cetirizine). Both first- and second-generation antihistamines are effective at controlling symptoms of AR [32].

Intranasal antihistamines (e.g., azelastine) are also available and are more efficacious and rapid in their symptomatic relief than oral antihistamines [3]. Anti-H3 and H4 antihistamines are currently under study for use in AR, but no agents have yet received approval [30,37].

**Table 1 biomedicines-10-02486-t001:** Summary of pharmacotherapy in AR. Summary of the discussed therapeutic agents, their main mechanism of action, and their main effects exerted on MCs and immune cells.

Agent	Mechanism of Action	Effect on MCs and Immune Cells
Antihistamines	Histamine receptor blockade	Inhibiting histamine’s effect
GCSs	Modifications of gene transcription leading to reduction in inflammatory mediators synthesis (i.e., IL1-8, TNFα, ***IFN***-***γ***, and GM-CSF) and controlling preonset activation of MCs and eosinophils	Attenuated recruitment and activation of MCs, eosinophils, and other immune cells
LTRAs	Blockade of leukotriene receptors	Attenuating immune cell activation by leukotrienes
CS	MC stabilization	Limiting inflammatory mediator release from MCs; attenuating the allergic inflammation
Omalizumab	Blockade of IgE antibodies	Limiting MC activation and degranulation
Dupilumab	Blockade of IL-4Rα subunit of IL-4 and IL-13 receptors	Limiting MC activation and degranulation
Allergen immunotherapy	Early desensitization of MCs, generation of regulatory lymphocytes responses, and regulation of IgE production	Reduction in MC and eosinophil activity

***Intranasal GCSs*** (i.e., beclomethasone, budesonide, fluticasone propionate, mometasone furoate, and triamcinolone acetonide) may also be used as a first-line symptomatic monotherapy, as an adjunct to intranasal antihistamine, or in combination with oral antihistamines in patients with mild, moderate, or severe symptoms. Studies have shown that intranasal GCSs are superior to antihistamines in effectively reducing nasal inflammation and improving mucosal pathology. Oral and injectable GCSs may cause significant systemic adverse effects; therefore, even though they have been shown to be effective in alleviating AR symptoms, their use is not recommended and routine use should be avoided [38,39].

Concerning the mechanism of action of GCSs in AR, they are primarily used due to their ability to suppress various stages of the allergic inflammatory reaction as a result of modifications in gene transcription [11]. Their main anti-inflammatory effect is elicited via a reduction in the synthesis of cytokines, including IL1-8, TNFα, interferon-gamma (IFN-gamma), and GM-CSF, which in turn attenuates the recruitment, survival, and activity of inflammatory immune cells. Although GCSs can decrease remodeling, they seem to have minimal effects on reversing the structural changes in the nasal mucosa [1]. In addition, treatment with intranasal GCSs has been shown to control the preonset activation of eosinophils and MCs present in AR [38].

***Leukotriene receptor antagonists (LTRAs)*** such as montelukast and zafirlukast block leukotriene receptors on immune cells, thus blocking activation of the latter by leukotrienes secreted from activated MCs and other inflammatory cells such as eosinophils. LTRAs were found to be beneficial in patients with AR and resulted in significant improvements in daytime nasal symptoms and the quality of life [40]. Moreover, they were found to effectively attenuate nasal obstruction and *rhinorrhea* by blocking leukotriene-activated inflammation in the nasal lavage fluids and airways [41]. In terms of symptomatic treatment of AR, LTRAs may be used as an adjunct to antihistamines in mild AR or moderate-to-severe AR in patients who do not tolerate GCSs.

***Cromolyn sodium (CS)*** is an MC stabilizer that prevents the subsequent release of inflammatory mediators including histamine and leukotrienes. It is an FDA-approved medication for adjunctive treatment of allergic rhinitis that is used in the form of nasal inhalation and is capable of effectively reduce sneezing, *rhinorrhea,* and nasal pruritus [42]. Recent in vivo and in vitro findings by our group demonstrated that immunomodulatory actions of CS could go beyond MC stabilization via a contribution to an increase in CD300a receptor levels and IL-10 cytokine release from IgE-activated MCs, thereby contributing to attenuation of the allergic inflammation [43].

***Omalizumab*** is a human monoclonal antibody designed to target and block IgE. It is capable of reducing free (and hence bound) IgE, downregulating high-affinity IgE receptors, and limiting MC degranulation, thus minimizing the release of mediators throughout the allergic inflammatory cascade. A meta-analysis that assessed the efficacy and safety of omalizumab in inadequately controlled AR in randomized controlled trials (RCTs) found that it reduced the Daily Nasal Symptom Severity Score (DNSSS), improved the Daily Ocular Symptom Severity Score (DOSSS), improved the Rhinoconjunctivitis Quality of Life Questionnaire, and reduced the mean daily consumption of antihistamines. In addition, in a recent study, long-term therapy with omalizumab was demonstrated to be effective and safe in treating severe persistent AR and concomitant asthma. These results supported the efficacy and safety of omalizumab in the management of patients suffering from inadequately controlled AR in conventional treatment [44,45]. Omalizumab has been approved by the FDA for the treatment of CRSwNP following the results of a phase III clinical trial that showed how omalizumab significantly improved clinical and endoscopic outcomes in severe CRSwNP with insufficient response to intranasal GCs [46]. However, omalizumab therapy is associated with significant costs, which is a limiting factor in its use as a treatment for AR [47].

***Dupilumab*** is a fully humanized monoclonal antibody that exerts its effects by targeting the interleukin IL-4Rα subunit of the IL-4 and IL-13 receptors, thus inhibiting the signaling of IL-4 and IL-13 (Figure 2). As explained earlier, these cytokines are key drivers of atopic diseases due to their role in regulating IgE production. Dupilumab was found to significantly improve AR-associated nasal symptoms in patients with uncontrolled persistent asthma and comorbid perennial AR (PAR) [48]. As for CRSwNP, two clinical studies (SINUS-24 and SINUS-52) evaluated dupilumab treatment in adult patients with severe uncontrolled CRSwNP. The results demonstrated that dupilumab was well tolerated and reduced polyp size, sinus opacification, and symptom severity [49]. Moreover, in a recent indirect treatment comparison of omalizumab to dupilumab use in CRSwNP, it was demonstrated that dupilumab demonstrated consistently greater improvements in key CRSwNP outcomes versus omalizumab at week 24 [50].

***Immunotherapy*** for patients in whom avoidance measures and combination pharmacotherapy are not effective should be considered [51]. Both innate and adaptive immune responses that contribute to allergic inflammation are suppressed by AIT [52] via early desensitization of MCs, generation of regulatory lymphocytes responses, regulation of IgE production, decreasing the activity of eosinophils and MCs, and generation of regulatory lymphocyte responses [53]. Allergen immunotherapy (AIT) is currently the only available disease-modifying therapeutic option. AIT can be administered via either the subcutaneous (SCIT) or the sublingual (SLIT) routes. Weekly incremental doses are given for 6 to 8 months followed by maintenance doses for 3 to 5 years. Typically, both routes of administration are safe and effective and are capable of leading to tolerance that lasts years after treatment cessation. Therefore, patients may experience a prolonged protective effect and other therapy may be ceased [51].

## 10. Conclusions and Outlook

Research in the last two decades has provided increasing evidence that MCs critically contribute to innate host defense and adaptive immunity. MCs play an important role in the initiation and progression of several rhinopathies and the primary role in AR. MCs initiate the allergic inflammation cascade that leads to the development and maintenance of AR through the secretion of inflammatory mediators and interplay with different inflammatory cells in the nasal mucosa. Available pharmacologic treatments that aim to modulate MCs and their mediators’ effects include GCSs, antihistamines, antileukotrienes, biologic agents such as dupilumab and omalizumab, and immunotherapy. Despite the current advancements and the introduction of biologics and immunotherapy, additional studies are required to reveal future therapeutic targets to attenuate MCs’ inflammatory activity in AR. Possible new targets may include sCD48, which has been found to be significantly higher in IAR patients during the pollen season. Other targets worthy of further investigation include proteins coded by differentially expressed genes in AR patients, especially tryptase (encoded by TPSAB1), a primary molecule in MCs’ preformed granules. Cytokines such as IL-33, IL-25, TSLP, and eotaxin should be further investigated as well due to their ability to promote continuous recruitment of eosinophils and T-helper cells to the inflamed tissue, which maintains the chronic inflammatory phase. Finally, further investigation of these ligands and the role of cytokines in AR should be carried out, as they may be potential novel targets for directed AR pharmacologic therapy or biomarkers for the evaluation of AR treatment regimens.

## Figures and Tables

**Figure 1 biomedicines-10-02486-f001:**
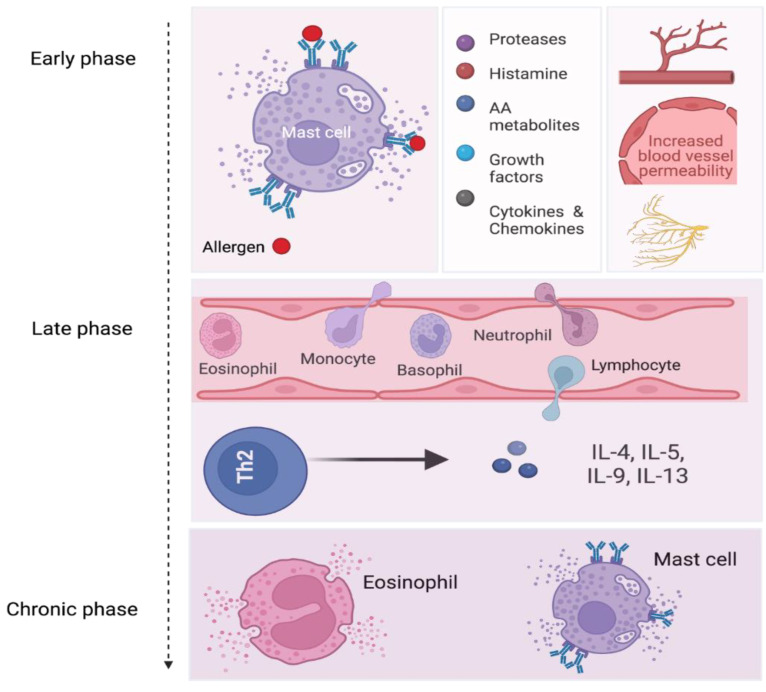
**MCs’ role in the pathophysiology of AR.** During the early phase of allergic inflammation, activated MCs secrete their preformed and de novo synthesized mediators, leading to nasal itching (by stimulating nerve endings), angiogenesis, vasodilation, and an increase in vascular permeability, then leading to nasal congestion and *rhinorrhea*. Cytokines and chemokines produced and released during the early phase induce the recruitment of various inflammatory cells to the site of inflammation. These cells secrete inflammatory mediators, further enhancing the inflammation and promoting its prolongation. MCs and eosinophils are predominantly found during the late and chronic phases of inflammation when resolution fails to occur. Figure 1 created using BioRender.com.

**Figure 2 biomedicines-10-02486-f002:**
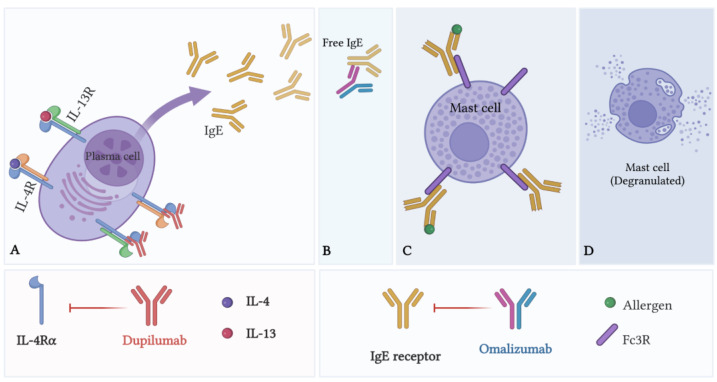
Dupilumab and omalizumab modulate MC activation. Dupilumab is a fully human monoclonal antibody that binds IL-4Rα. It blocks IL-4 and IL-13 signaling, thus inhibiting IgE production (**A**). Omalizumab is a humanized monoclonal antibody that binds free IgE in the blood (**B**), preventing IgE from binding to MC-bound Fc3R (**C**), thus inhibiting IgE-dependent MC activation (**D**). Figure 2 created using BioRender.com).

## Data Availability

Not applicable.

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
