# Peer review of "Allergic Rhinitis: Pathophysiology and Treatment Focusing on Mast Cells"

_biomedicines, 2022, doi:10.3390/biomedicines10102486_

Round 1
Reviewer 1 Report
A review article on the role of mast cells in allergic rhinitis. Generally, a well written article. But need major changes to make it consistent and relevant.
As the authors main aim was to update on the pathophysiology and mechanism of action of mast cells, I am not clear which part of the manuscript covered this section. Is there any new update at all? Or the manuscript just did another review of an already established concept in allergic rhinitis pathway?
The introduction stated the coexistence of CRS among others with AR. However, there is no explanation and further elaboration on their link. It is now acknowledged that the phenotype of CRS includes the allergic CRS type in addition to eosinophilic CRS as a type 2 CRS. I suggest the authors to focus on AR as this is their main topic and leave out CRS. The introduction should cover an in depth review of AR and how understanding of the mast cells role can help us to treat it better.
It would also be useful if the authors can add a topic on the interaction of mast cells with the other inflammatory mediators in AR and AR with comorbid allergic conditions such as bronchial asthma and atopic dermatitis. Certainly this would make the review an interesting reading.
In subtopic no 6, the authors covered both AR and CRS as if they are the same. I would suggest the authors to remove CRS and just focus on AR and other well-established comorbid allergic conditions like asthma and atopic dermatitis.
In subtopic no 9 where AR pharmacotherapy being discussed, the addition of a table highlighting the mechanism of action of each agent and its effect on the mast cell and inflammatory cells will make the section easier to read.
Author Response
Response to Reviewer 1
Point 1: As the authors’ main aim was to update on the pathophysiology and mechanism of action of mast cells, I am not clear which part of the manuscript covered this section. Is there any new update at all? Or the manuscript just did another review of an already established concept in allergic rhinitis pathway?
Response 1: In this review our aim is to provide a comprehensive summary of MCs role in AR, as well as any available update on MCs (for example, please refer to subtopic 7). However, we do agree that for the most part there are no updates, therefore, and in order to be more specific, we have changed the title to: Allergic Rhinitis: Pathophysiology and Treatment Focusing on Mast Cells.
Point 2: The introduction stated the coexistence of CRS among others with AR. However, there is no explanation and further elaboration on their link. It is now acknowledged that the phenotype of CRS includes the allergic CRS type in addition to eosinophilic CRS as a type 2 CRS. I suggest the authors to focus on AR as this is their main topic and leave out CRS. The introduction should cover an in depth review of AR and how understanding of the mast cells role can help us to treat it better.
Response 2: In this review we are focusing on MCs role in AR. The symptoms of AR may overlap with rhinosinusitis. Both AR and CRSwNP (the most dominant type) share a common etiopathological background of a Th2 driven inflammation. Therefore, novel pharmacologic approaches currently used to treat CRSwNP may be relevant to developments in AR, as recent findings would suggest. Thus we do believe it is relevant to keep information in the introduction on CRSwNP, information regarding CD48 & CRSwNP, Dupilumab and CRSwNP and the connection to AR. To be clearer and specific on the link between these two entities, we have added to the introduction how Th2 type inflammation links CRSwNP and AR. Relevant information to the scope of this review, regarding CRS endotyping, is also included in the introduction which was expanded to include more details about AR relevant to the scope of the review, and to MCs’ role in AR (which is discussed in detail later in the review).
Point 3: It would also be useful if the authors can add a topic on the interaction of mast cells with the other inflammatory mediators in AR and AR with comorbid allergic conditions such as bronchial asthma and atopic dermatitis. Certainly this would make the review an interesting reading.
Response 3: We understand that the reviewer would like a topic on MCs interaction with other mediators in comorbid conditions, including bronchial asthma and atopic dermatitis, to be added, and we agree that it is a very important topic. Still, we prefer not to deviate from the scope of this review, which is to comprehensively and specifically summarize MCs involvement in the development of AR symptoms. We briefly added a paragraph on MCs interaction with eosinophils in the AEU in allergic inflammation including AD and allergic asthma in the introduction. MCs interactions with other inflammatory mediators and cells, in AR, is elaborated in sections 2-5.
Point 4: In subtopic no 6, the authors covered both AR and CRS as if they are the same. I would suggest the authors to remove CRS and just focus on AR and other well established comorbid allergic conditions like asthma and atopic dermatitis.
Response 4: The paragraph (subtopic 6) states clearly what data is pertinent to CRSwNP and what is pertinent to AR. We do believe it is of added value to mention the finding related to CD48 in CRSwNP due to potential overlap in clinical treatment and further development. AR and CRSwNP are both rhinopathies that may present with similar symptoms, while asthma and AD are not, rendering them out of the scope of this review in our opinion.
Point 5: In subtopic no 9 where AR pharmacotherapy being discussed, the addition of a table highlighting the mechanism of action of each agent and its effect on the mast cell and inflammatory cells will make the section easier to read.
Response 5: Table 1, briefly summarizing the main points discussed in subtopic 9, has been added based on your suggestion.
Thank you for your valuable comments.
Reviewer 2 Report
1-Please type the Latin terms in italic style.
2.-Please clarify this sentence "In addition, adequate asthma control is compromised by uncontrolled AR."
3.-I suggest introducing a paragraph mentioning Aspirin-Exacerbated Respiratory Disease; and the role of mast cells in this disease.
Author Response
Point 1: Please type the latin terms in italic style.
Response 1: We made this change in the manuscript as requested.
Point 2: Please clarify the sentence "In addition, adequate asthma control is compromised by uncontrolled AR".
Response 2: We changed and revised the introduction section. In the revised version this sentence was deleted.
Point 3: I suggest introducing a paragraph mentioning Aspirin-Exacerbated Respiratory Disease; and the role of mast cells in this disease.
Response 3: A paragraph mentioning AERD has been added to the introduction.
Thank you for your valuable comments.
Reviewer 3 Report
I have thoroughly enjoyed reading this article. It is already well known and described that mast cells have an outstanding role in allergic rhinitis.
Nevertheless, a Review article on the topic so elegantly outlined, rich of information and well selected references was required.
The authors have already contributed with scientific papers pertinent the topic and therefore they cite themselves appropriately. The references cited are the most relevant in the field.
The Figures are highly explicative and well designed and easy to consult.
I deeply approve of the manuscript and suggest its publication on Biomedicines.
Author Response
We would like to thank you for your review.
No comments to be addressed.
Round 2
Reviewer 1 Report
Concerns have been addressed. Thank you.